



# Quality assessment of Dobson spectrophotometers for ozone column measurements before and after automation at Arosa and Davos

René Stübi[1], Herbert Schill[2], Eliane Maillard Barras[1], Jörg Klausen[1], and Alexander Haefele[1]

[1]Federal Office of Meteorology and Climatology, MeteoSwiss, 1530 Payerne, Switzerland
[2]Physikalisch-Meteorologisches Observatorium / World Radiation Center, 7260 Davos Dorf, Switzerland

*Correspondence to:* R. Stübi (rene.stubi@meteoswiss.ch)

**Abstract.** The longest ozone column measurements series are based on the Dobson sun spectrophotometers developed in the 1920s by Prof. G. B. W. Dobson. These instruments still constitute an important part of the World Meteorological Organization's global network due to their optical qualities and ruggedness. The primary drawback of this instrument is the effort needed for its manual operation. In industrialized and some lesser developed countries, most stations have made the choice to replace the Dobson by the automated Brewer sun spectrophotometers but some are still relying on the Dobson instrument. One of them is the Arosa station where both instrument types are run in parallel. Here, an automated version of the Dobson instrument was developed and implemented recently. In the present paper, the results of the analysis of simultaneous measurements from pairs of Dobson instruments that were either collocated at Arosa or Davos, or operated one at each location, are presented for four distinct time periods:

– 1992–2012 : Manual vs. Manual operation of Collocated Dobson instruments (MMC)

– 2012–2013 : Manual vs. Automated operation of Collocated Dobson instruments (MAC)

– 2012–2019 : Automated vs. Automated operation of Collocated Dobson instruments (AAC) and

– 2016–2019 : Automated vs. Automated operation of Distant Dobson instruments (AAD)

The direct comparison of two instruments using the standard operation procedure during the MMC period gives a metric necessary to validate the automated version of Dobson instruments. The direct comparison of two collocated instruments using the standard manual operation procedure reveals random differences of coincident observations with a standard deviation of ∼0.45 % and monthly mean differences between -1.0 and +0.8 %. In most cases the observed biases are not statistically significant. The same analysis of two automated Dobson instruments yields significantly smaller standard deviation of ∼0.25 % and biases of between -0.7 % and 0.8 %. This demonstrates that the repeatability has improved with the automation while the systematic differences are only marginally smaller.

The description of the automated data acquisition and control of the Dobson instrument is presented in a separate paper (*Stübi et al.* , 2020).



# 1 Introduction

In 2017, the celebration of 30 years of Montreal protocol (*Albrecht and Parker*, 2019) was a reminder of this important worldwide agreement to ban the use of substances harmful for the ozone layer in industrial processes as well as their release to the atmosphere (*Solomon*, 1999). Present monitoring activities show the effectiveness of the protocol in the stabilisation

and the decrease of their abundance in the atmosphere. However, while it seems to have stabilized since the beginning of the $21^{st}$ century, the expected recovery of the ozone layer to the pre-1980 level has still not been observed in most parts of the atmosphere. Hence it remains important to continue with the monitoring at global scale. The anxiety about the ozone hole has favoured the development of well organised dedicated monitoring networks based in particular on the Dobson and Brewer instruments. In these networks, the "Light Climatic Observatory" (LKO for german "LichtKlimatsches Observatorium") at

Arosa has a special renown since it provides the longest continuous total ozone column measurement series as illustrated in Figure 1. The ozone column decline of the 1970s-1980s is clearly seen, followed by a leveling off since the beginning of the $21^{st}$ century. However no sign of the expected recovery of the ozone layer is present up to now in the LKO total column ozone time series nor at other ground based stations (*Ball et al.* , 2019). The slight decrease of the variability illustrated by the shading is probably dominated by the measuring technique improvement over the decades. The trend analysis of the ozone

abundance at different altitude ranges is still the subject of research and publications (e.g *Pawson et al.* (2014); *WMO* (2018); *SPARC/IO3C/GAW* (2019)). A clear sign of recovery is presently observed at mid-latitude high altitude (∼40 km) in accordance with numerical models forecast. This is interpreted as a positive consequence of the Montreal protocol. In contrast, another recent publication (*Ball et al.* , 2018) still revealed a negative ozone trend in the lower stratosphere (∼10 km) and reminded the public that the rhetoric of the ozone problem being already solved was overly optimistic.

The history of the LKO and the essential role of ozone pioneers in keeping a measurement site active over such a long period of time was detailed in two recent publications (*Staehelin et al.* , 2018; *Staehelin and Viatte*, 2019). The link between the LKO activities and societal concerns was highlighted in particular with the tuberculosis treatment in the earlier years and the ozone hole more recently. The LKO ozone column measurements series and the succession of instruments in operation has been analysed in (*Perl and Dütsch*, 1958; *Dütsch*, 1984; *Brönnimann et al.*, 2003). *Staehelin et al.* (1998) described 4 decades of

use of Dobson D015 at Arosa from 1948 until 1992 and the arrival of Dobson D101 in 1966 as a redundant instrument. With the decommissioning of D015 in 1992, D101 became the reference instrument and the newly arrived Dobson D062 took the role of the redundant instrument. The instruments were upgraded with a digital recording of the R-dial position at the end of the 1980s (*Hoegger et al.* , 1992) but continued to be manually operated in a dedicated convenient-to-use rotating cabin.

Prof. Dütsch, responsible scientists for LKO, made first attempts to automate the Dobson instrument in the 1970s (*Räber*,

1973). For technical reasons, the project was suspended for the direct sun measurements but was continued for the zenith measurements (Umkehr). The implementation of a fully automated version of the Dobson instrument developed at MeteoSwiss between 2012 and 2014 motivated the new analysis of the data as presented here. More technical aspects of the automation are described in a separate paper (*Stübi et al.* , 2020). The automated Dobson instruments require only occasional presence on site essentially for lamps tests. Following this transition to automated operation, the comparison of the two sites Arosa and Davos





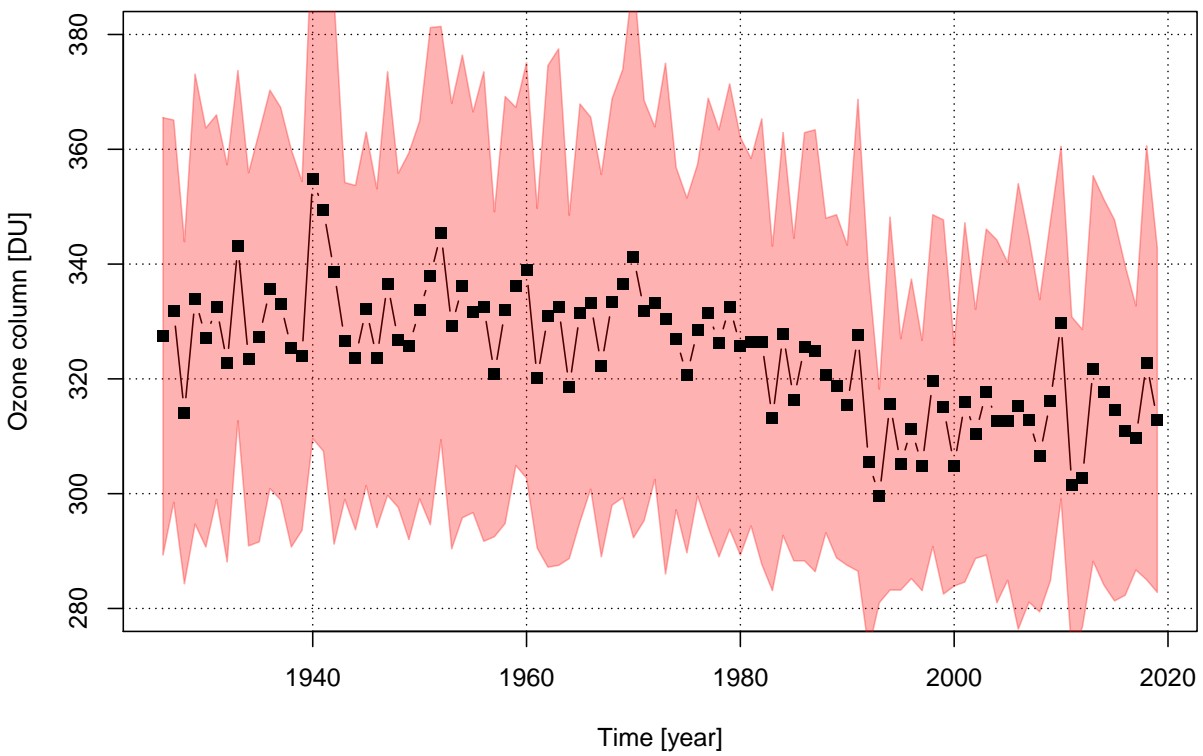

**Figure 1.** Arosa ozone column time series : yearly mean values in Dobson Units [DU]. The shading are calculated as the standard deviation of the monthly means of each year.

started, with a view to continue the world's longest total column ozone series based on Dobson observations in Davos.

The present study is centered on the analysis of Dobson instruments data and is a follow-up of two previous analyses of the LKO Brewer triad measurements (*Stübi et al.*, 2017a, b).

The paper is organized as follows: in section 2, the measurement principles are presented, followed in section 3 by a description

5 of the data sets and of the data quality control procedures applied. The results of the analysis are presented in section 4, and the discussion of the results in section 5.

## 2 Dobson spectrophotometer measurements

The principle of the Dobson instrument is described in many publications (*Dobson*, 1968; *Komyhr*, 1980; *Evans*, 2008; *Scarnato et al.*, 2009, 2010; *Moeini et al.*, 2019). The intensity of the sun's radiation in the UV range at ground level is modulated

10 by the amount of ozone in the atmosphere. The sun spectrophotometers of type Dobson and Brewer measure the intensity at a few specific wavelengths in the range 310–340 nm. In the Dobson instrument, the sun light is diffracted by a prism and two narrow slits allow to select the different pairs of wavelength commonly referred to as A (305.5 nm / 325.4 nm), C (311.45 nm



/ 332.4 nm) and D (317.6 nm / 339.8 nm). These pairs are combined to form the double pairs AD and CD used to calculate the ozone column while eliminating atmospheric interferences (*Evans*, 2008; *Basher*, 1982). Following *Evans* (2008) notation, the ozone column is retrieved with the following formula:

$$O_3 = X_{AD} = \frac{(N_A - N_D) - [(\beta^s - \beta^l)_A - (\beta^s - \beta^l)_D]\frac{mp}{p_0} - [(\delta^s - \delta^l)_A - (\delta^s - \delta^l)_D]sec(SZA)}{[(\alpha^s - \alpha^l)_A - (\alpha^s - \alpha^l)_D]\mu} \tag{1}$$

where the superscripts s (l) refer to the short (long) wavelength within each pair, $\alpha^\lambda$ is the absorption coefficient of ozone, $\beta^\lambda$ and $\delta^\lambda$ are respectively the Rayleigh and Mie scattering coefficients, m and $\mu$ refer to the air masses for Rayleigh and ozone respectively. The ratio $p/p_0$ is a correction for the mean station pressure and SZA is the solar zenith angle. The measured N values are the differences of the solar radiation intensity ratios $I_0^s/I_0^l$ at the top of the atmosphere and $I^s/I^l$ at the surface:

$$N_A - N_D = [log(\frac{I_0^s}{I_0^l}) - log(\frac{I^s}{I^l})]_A - [log(\frac{I_0^s}{I_0^l}) - log(\frac{I^s}{I^l})]_D \tag{2}$$

The wavelength dependence of the Mie scattering is much smaller than the dependence of ozone and Raleigh scatterings therefore the last term of equation 1 is negligible for the double pairs. In the Brewer instruments, a diffraction grating selects four wavelengths (310.1 nm, 313.5 nm, 316.8 nm, 320.0 nm) which are then combined in a similar way as for the Dobson instrument to extract the ozone column (*Kerr et al.*, 1981; *Kerr and McElroy*, 1995). The Dobson ozone column retrieval algorithm is fairly simple and assumes similar characteristics for all instruments, characteristics based on the optical properties of the primary

reference Dobson instrument $D_{083}$ (*Komhyr et al.*, 1989). In the past 10 years, the EMRP-ATMOZ project has contributed to an improved understanding of the sun spectrophotometer's measurement principle (*ATMOZ*, 2018). Thus, measurements of the Dobson slit functions (*Köhler et al.*, 2018), of the ozone cross-sections and their temperature dependencies (*Bass and Paur*, 1985; *Serdyuchenko et al.*, 2014; *Malicet et al.*, 1995; *Janssen et al.*, 2018), of the stray-light effect (*Christodoulakis et al.*, 2015; *Karppinen et al.*, 2015; *Moeini et al.*, 2019) and their implications on the ozone column retrieval for different instruments

(*Redondas et al.*, 2014) are now available. An adaptation of the processing algorithm with these recent findings would certainly improve the absolute accuracy of the ozone observations. However, it is impossible to apply these findings consistently to the historical records of Dobson measurements because some essential instrument characteristics (slit functions, wavelengths in use, etc.) are not available for older instruments and data sets.

The Dobson network calibration is organised by the World Meteorological Organization's (WMO) Global Atmosphere Watch

program. It is based on absolute calibration of a primary reference and six regional secondary traveling standards to transfer the primary reference scale to each individual station (*Komhyr et al.*, 1989). These calibrations were carried out regularly at LKO as indicated in Figure 3 by the arrows.

The Dobson automation and re-location from LKO to the Physikalisch-Meteorologisches Observatorium Davos / World Radiation Center (PMOD/WRC) was considered by MeteoSwiss with the prospect of perpetuating the measurements in the long

term under optimal conditions. Factors considered in the analysis included the availability of operators for a year-round 24/7 monitoring program, data quality improvements (repeatability, reproducibility, increased frequency of measurements) and reduction of operational cost due to institutional synergies. Great care was taken to avoid a fundamental change of the Dobson measurements and hence to support the continuity of the LKO ozone column time series.





**Table 1.** Chronology of the interventions and calibrations of the three Dobson instruments.

| Date | Dobson Instruments | Comment |
|---|---|---|
| 1992 | $D_{062}$ | 062 put in operation |
| 01.09.2001 | | New station manager / operator (R. Burren) |
| 12.07.2010 | $D_{051}$, $D_{062}$, $D_{101}$ | 12.-16.07.2010 Inter-comparison (D074 of SOOH) |
| 21.03.2011 | $D_{051}$ | 21.-31.03.2011 New electronics (Payerne) |
| 01.05.2011 | | New station manager / operator (W. Siegrist) |
| 13.10.2011 | $D_{062}$ | New photo-multiplier amplifier board |
| 18.01.2012 | $D_{051}$ | 18.01.-15.02.2012 workshop Payerne (automation) |
| 13.03.2012 | $D_{062}$ | 13.03.-11.04.2012 workshop Payerne (automation) |
| 16.07.2012 | $D_{051}$, $D_{062}$, $D_{101}$ | 16.-27.07.2012 Inter-comparison (D064 of MOHp) |
| 15.11.2012 | $D_{062}$ | Change of sun-director prism (R-values shift by $\sim$5 units) |
| 04.03.2013 | $D_{051}$ | Begin of total ozone measurements |
| 09.11.2013 | $D_{101}$ | 09.11.13-18.05.14 workshop Payerne (automation) |
| 18.05.2014 | $D_{051}$, $D_{101}$ | Double container as new Dobson housing |
| 03.07.2014 | $D_{101}$ | Restart with automated system |
| 21.07.2014 | $D_{051}$ | New amplifier board (discontinuity in SL-values) |
| 02.2015 | $D_{051}$, $D_{062}$ | New quartz dome |
| 07.2015 | $D_{051}$, $D_{062}$, $D_{101}$ | New azimuth control of the turntables + $D_{101}$ new quartz dome |

A description of the technical details of the automated system is found in a separate publication (*Stübi et al.*, 2020). Table 1 lists the dates of the main changes that have the potential to introduce changes in the measurements of the three LKO Dobson instruments. By the end of 2015, all three Dobson instruments were automated and had reached the same configuration.

## 3 Data sets of coincident measurements

The automation of the Dobson spectrophotometers $D_{062}$ and $D_{051}$ was performed between the Inter-comparisons of summer 2010 and summer 2012, while $D_{101}$ was automated at the beginning of 2014. Until early 2016, the three Dobson instruments were at LKO as illustrated by the red and blue color bars in the upper panel of Figure 2. Then, Dobson $D_{101}$ was moved to the PMOD/WRC. *Stübi et al.* (2017b) have described the stations and have analysed the similar re-location from Arosa to Davos of Brewer instruments in terms of differing environmental factors with a potential to break the LKO ozone column series.

The lower 3 panels of Figure 2 show the last 10 year of the standard lamps corrections applied for the AD pairs for the three Dobson instruments. A variation of the difference $\delta N_A$-$\delta N_D$ of 0.5 corresponds to $\simeq$1% of the ozone column variation at air mass $\mu = 1$, decreasing as $1/\mu$ that is $\simeq$0.5% at $\mu = 2$. These panels illustrate the stability of the instruments resulting from regular lamp tests and the adjustments from the maintenance / calibration campaigns (yellow lines). Dobson $D_{101}$ drifted slowly between 2010–2018 while $D_{051}$ and $D_{062}$ were particularly stable besides the 2011 increase of $D_{062}$. The weather

**Figure 2.** Upper panel: historical changes of the Dobson operation as well as of the instrument locations. Lower three panels : time series 2010–2020 of the lamp corrections $\delta N_A - \delta N_D$ for the three LKO Dobson instruments. The vertical yellow bars denote the intercomparisons with the European travelling standard. The blue bands mark the unavailability of each Dobson instrument during the process of automation.



during the 2017 calibration campaign was not fair enough for a good evaluation of the calibration status of the LKO Dobson instruments. Therefore the 2017 calibration is not taken into account in this study.

Since the Dobson Inter-comparison in July 2012, $D_{051}$ (previously dedicated to ozone profile measurements with the Umkehr method) has also been used for total ozone measurements. Regular $D_{051}$ direct sun measurements began however only in March

2013. Therefore the overlap with the reference instrument $D_{101}$ lasted only 8 months before $D_{101}$ went to the workshop for the automation.

For the present analysis, measurements between a pair of Dobson instruments were defined as coincident if the following criteria were met : time difference $\delta t < 300$ seconds, air mass difference $\delta\mu < 0.05$ and air mass $\mu \leq 4$. At LKO, the manual operation was facilitated by having the two instruments side-by-side on a turntable, which resulted in a systematic time

difference $\delta$t between 45 and 75 seconds. For the automated operation, the mean $\delta$t is close to zero seconds.

### 3.1 Data quality control

Until end of 2011, all manual measurements underwent a data quality control on a daily basis. The individual measurements were flagged based on a visual comparison of all Dobson (AD, CD wavelengths double pairs) and Brewer instruments. The meteorological parameters (e.g global radiation, sunshine duration and rain) were also considered in this process. This approach

involved subjective flagging by an experienced scientist. With the increase of the number of measurements by a factor $\sim$10 following the automation in the course of 2012, a different approach was developed. Since then, each Dobson instrument has been treated separately for the single wavelength pairs C, D, A and for the double pairs AD and CD. In a first step, the sun duration for 10-min periods is used as additional information, measurements in periods with less than 4 minutes of sun are flagged. Then the standard deviation of the 20 seconds R-dial records ($\delta$R) is used as a quality criterion. In the next

step, an algorithm based on consecutive elimination of bad or doubtful measurements is applied for flagging. A $4^{th}$ order polynomial function of time is calculated as a proxy of the daily variation. Outliers are eliminated (flagged) one by one, the polynomial function being recalculated after each elimination until all measurements of a day fulfill the wavelength and instrument dependant empirically determined criteria (e.g for $D_{062}$ $|$poly–$O_3| < 0.8\%$, $< 2.0\%$ and $< 1.0\%$ for respectively the C, D and A pair). The two minutes measurement cycle that was adopted helps to identify these outliers based on the assumption

that the total ozone abundance changes slowly over time. Therefore, two consecutive measurements must also agree within a given limit. Once these limits and convergence criteria are established, the flagging is done automatically without human intervention. However the measurements of the different instruments are still compared by visual control in order to detect malfunctions or drifts in an individual Dobson, which would then be flagged manually.

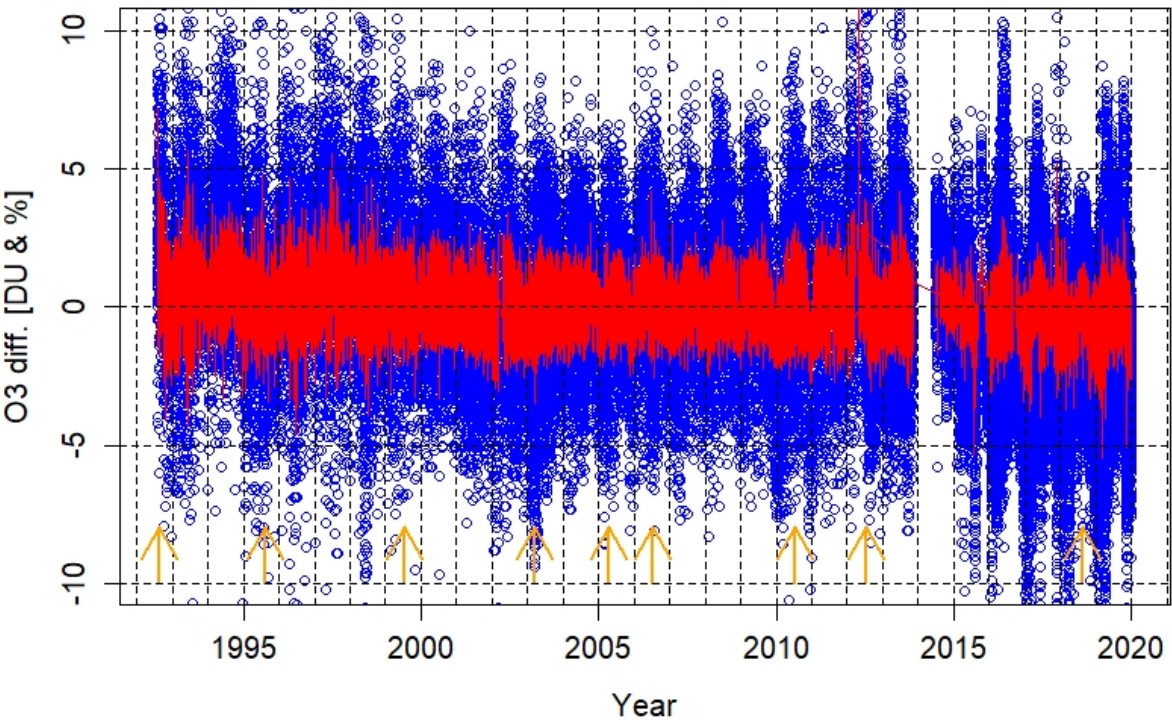

**Figure 3.** Time series of the differences between individual coincident measurements of the Dobson instruments $D_{101}$ and $D_{062}$ over the period 1992–2019. Blue: difference $D_{062}$ - $D_{101}$ in Dobson units [DU]. Red: difference $D_{062}$ - $D_{101}$ / $D_{101}$ in [%]. Yellow arrows indicate the calibration / maintenance campaigns.

## 4   Results

In Figure 3, the time series 1992–2019 of the differences between coincident measurements from Dobson $D_{101}$ and $D_{062}$ is shown. We can observe the generally good agreement between these two sets of independent measurements. The transition period 2011–2014 from manual to automated Dobson operations stands out with larger discrepancies and variability in the differences due to the adaptations of the data acquisition system and of the measuring program. Regular calibration campaigns in conformity with the Dobson network procedures which state a 4–5 years calibration cycle are marked on Figure 3 by arrows.

Different time periods are considered defined by the type of operation (manual vs. automated) and the location of the instruments (Arosa vs. Davos):

-   1992 - 2012 : Manual vs. Manual operation of Collocated Dobson instruments (MMC) at LKO

-   2012 - 2013 : Manual vs. Automated operation of Collocated Dobson instruments (MAC) at LKO

-   2013 - 2019 : Automated vs. Automated operation of Collocated Dobson instruments (AAC) at LKO or Davos





**Table 2.** Median differences of coincident data averaged over the whole data set for the four sub-periods. $P_{2.5\%}$, resp. $P_{97.5\%}$, are the percentiles 2.5%, resp. 97.5% of the sample and IPR is the range between them.

| Time Period | Reference Dobson | Redundant Dobson | Type | Sample size | Difference [%] | | | IPR |
| --- | --- | --- | --- | --- | --- | --- | --- | --- |
| | | | | | $P_{2.5\%}$ | Median | $P_{97.5\%}$ | $2.5\%-97.5\%$ |
| 1992–2012 | $D_{101}$ (LKO) | $D_{062}$ (LKO) | MMC | 31129 | -1.67 | 0.14 | 2.08 | 3.75 |
| 2012–2013 | $D_{101}$ (LKO) | $D_{062}$ (LKO) | MAC | 1907 | -1.79 | -0.07 | 2.06 | 3.85 |
| 2013–2013 | $D_{101}$ (LKO) | $D_{051}$ (LKO) | MAC | 627 | -1.25 | 0.62 | 2.54 | 3.79 |
| 2014–2016 | $D_{101}$ (LKO) | $D_{062}$ (LKO) | AAC | 22247 | -1.17 | -0.03 | 1.11 | 2.28 |
| 2014–2016 | $D_{101}$ (LKO) | $D_{051}$ (LKO) | AAC | 7195 | -1.69 | 0.25 | 1.47 | 3.16 |
| 2014–2018 | $D_{062}$ (LKO) | $D_{051}$ (LKO) | AAC | 41134 | -0.98 | 0.00 | 0.87 | 1.85 |
| 2018–2019 | $D_{101}$ (DAV) | $D_{051}$ (DAV) | AAC | 4531 | -0.40 | 0.13 | 0.81 | 1.21 |
| 2016–2019 | $D_{101}$ (DAV) | $D_{062}$ (LKO) | AAD | 48957 | -1.80 | -0.44 | 1.11 | 2.91 |
| 2016–2019 | $D_{101}$ (DAV) | $D_{051}$ (LKO) | AAD | 20471 | -1.72 | -0.47 | 0.98 | 2.70 |
| 2018–2019 | $D_{051}$ (DAV) | $D_{062}$ (LKO) | AAD | 3221 | -1.56 | -0.38 | 0.98 | 2.54 |

– since 2016 and planned until 2021 : Automated vs. Automated operation of Distant Dobson instruments (AAD)

Table 2 shows the statistics of the observed differences for these different periods of operation of the Dobson instruments. Since there are 3 instruments and 2 locations, different cases for a given period are present in Table 2. In the MMC 20 year period, only $D_{101}$ and $D_{062}$ were used for total ozone measurements. The median difference is 0.14% with a 2.5%–97.5% inter-

percentile range ($IPR_{2.5\%-97.5\%}$) slightly below 4%. The two instruments were in very good agreement with no significant difference. Considering an average of 250 sunny days a year in Arosa, the 31'129 data points correspond to 6 to 7 coincident observations per day. On the relatively short MAC transition period, automated $D_{062}$ and $D_{051}$ were compared to manual $D_{101}$. For the pair $D_{101}/D_{062}$ the results are very similar to the MMC case, but for the pair $D_{101}/D_{051}$ a non significant bias of $\sim$0.6% is observed. The AAC comparison period shows an increase of the sample sizes by a factor of $\sim$10 together with a reduced

$IPR_{2.5\%-97.5\%}$ and no significant differences. Finally for the AAD period, an intermediate $IPR_{2.5\%-97.5\%}$ and a non significant bias $\sim$ 0.4% were found.

In *Stübi et al.* (2017a), an analysis of the daily Brewer data to discern the mid to long term variations of the differences and the short term random fluctuations of coincident measurements was introduced. This was an alternative method to the one introduced by *Fioletov et al.* (2005) to study the stability of the Toronto Brewer reference triad. Recently *León-Luis et al.*

(2018) published an analysis of the Izaña Brewer triad using both approaches and they concluded that results are similar for the two analysis methods. As illustrated in Figure 4, the analysis of *Stübi et al.* (2017a) involves fitting one single $4^{th}$ order polynomial function of time to both sets of measurements for the day considered. This function simulates the mean behavior of the ozone column during that day. For each instrument two parameters are calculated : first, the bias $\delta$ (in [DU]) between the polynomial function and the data subset and second, the standard deviation $\sigma$ of the measurements around the fit. The

difference of the two $\delta_i$, $\Delta_{12} = \delta_1 - \delta_2$, corresponds to the mean bias between the two instruments for that day and is positive





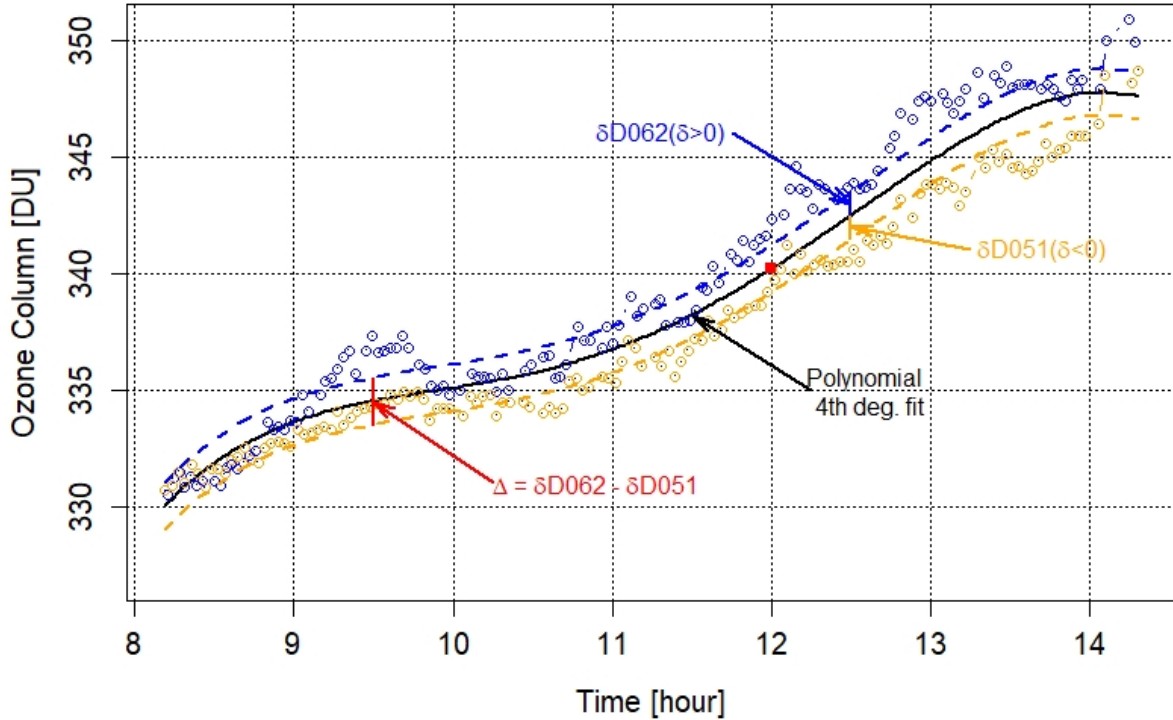

**Figure 4.** Illustration of the daily values of coincident $D_{062}$ and $D_{051}$ automated Dobson measurements for March 27 2016. The black line is the polynomial fit and the dashed lines correspond to the bias $\delta_{062}$ of $D_{062}$ data (blue), respectively the bias $\delta_{051}$ of $D_{051}$ data (orange). The difference, $\Delta_{062-051}$ (red), is the bias between $D_{062}$ and $D_{051}$ instruments evaluated from the coincident measurements of that day.

if values from instrument 1 are larger than those of instrument 2 (see Figure 4). $\sigma_i$ is a measure of the random fluctuations of each instrument, i.e., its repeatability. This approach works best with the numerous daily data available from the automated system but it can also be applied to the manual operation. The results of the daily analysis for the different periods mentioned above are presented in the next subsections.



**Table 3.** Mean monthly medians of parameters $\Delta_{ij}$ for the four sub-periods.

| Time Period | Reference | Redundant | Type | Months | $\Delta_{Ref-Red}$ [%] | | | IPR |
| --- | --- | --- | --- | --- | --- | --- | --- | --- |
| | | | | | $P_{2.5\%}$ | Median | $P_{97.5\%}$ | $_{2.5\%-97.5\%}$ |
| 1992–2012 | $D_{101}$ (LKO) | $D_{062}$ (LKO) | MMC | 234 | -0.95 | -0.13 | 0.77 | 1.72 |
| 2012–2013 | $D_{101}$ (LKO) | $D_{062}$ (LKO) | MAC | 16 | -1.05 | 0.19 | 0.55 | 1.60 |
| 2013–2013 | $D_{101}$ (LKO) | $D_{051}$ (LKO) | MAC | 9 | -1.68 | -0.56 | -0.11 | 1.57 |
| 2014–2016[1] | $D_{101}$ (LKO) | $D_{062}$ (LKO) | AAC | 25 | -0.59 | -0.03 | 0.78 | 1.37 |
| 2014–2016[1] | $D_{101}$ (LKO) | $D_{051}$ (LKO) | AAC | 9 | -0.23 | 0.45 | 0.79 | 1.02 |
| 2014–2018 | $D_{051}$ (LKO) | $D_{062}$ (LKO) | AAC | 46 | -0.68 | 0.00 | 0.60 | 1.28 |
| 2018–2019 | $D_{101}$ (DAV) | $D_{051}$ (DAV) | AAC | 8 | -0.22 | -0.13 | 0.08 | 0.30 |
| 2014–2019 | $D_{101}$ (DAV) | $D_{062}$ (LKO) | AAD | 46 | -0.30 | 0.53 | 1.05 | 1.35 |
| 2014–2019 | $D_{101}$ (DAV) | $D_{051}$ (LKO) | AAD | 29 | -0.22 | 0.50 | 1.09 | 1.31 |
| 2018–2019 | $D_{051}$ (DAV) | $D_{062}$ (LKO) | AAD | 9 | -0.05 | 0.43 | 1.18 | 1.23 |

[1] with additional two summer periods in 2017 and 2018

## 4.1 Period of manual operation 1992–2012 (MMC period)

Figure 5 illustrates the results of the daily analysis applied to the period 1992–2012 for the coincident Dobson $D_{101}$ and $D_{062}$ data. In the earlier years of parallel measurements, Dobson $D_{062}$ was between 0.5 and 1% higher than $D_{101}$ but this bias has gradually decreased and the two data sets have agreed within ±0.5% since about the year 2000. We note a shallow seasonal

5  cycle in the difference since 2005. The regular maintenance/calibration campaigns (black lines) did not induce noticeable breaks in the time series of differences. We also observe that the differences during the periods following calibrations are not always zero as expected. This is because each instrument was calibrated independently against the traveling standard and differences of ±0.5% are within the uncertainty of the calibration procedure itself and were therefore not compensated. The repeatability $\sigma_i$ is shown separately in the lower panle of Figure 5. Values between 0.3% to 0.6% were observed for both

10  instruments. The MMC section of Table 3, resp. of Table 4 summarize the statistics of the parameters $\Delta$, resp. $\sigma$ resulting from the daily analysis. The mean monthly median differences $\Delta$ are not significantly different from zero and the $IPR_{2.5\%-97.5\%}$ is 1.7%. The repeatability around $\sim$0.4% (0.3%–0.7%) for these two manually operated Dobson instruments is similar and probably varied depending on the operator's experience and skill. These numbers are our reference metrics for comparing results of manual and automated observations of the Dobson instruments in the next sections.





**Table 4.** Mean monthly median of the parameters $\sigma_i$ for the four sub-periods.

| Time Period | Instrument | Type | Months | $\sigma$ [%] | | | IPR | Remark |
|---|---|---|---|---|---|---|---|---|
| | | | | $P_{2.5\%}$ | Median | $P_{97.5\%}$ | $2.5\%-97.5\%$ | |
| 1992–2012 | $D_{101}$ (LKO) | MMC | 234 | 0.29 | 0.45 | 0.71 | 0.42 | vs. $D_{062}$ |
| 1992–2012 | $D_{062}$ (LKO) | MMC | 234 | 0.31 | 0.43 | 0.60 | 0.29 | vs. $D_{101}$ |
| 2012–2013 | $D_{101}$ (LKO) | MAC | 16 | 0.32 | 0.47 | 0.61 | 0.29 | vs. $D_{062}$ |
| 2012–2013 | $D_{062}$ (LKO) | MAC | 16 | 0.24 | 0.28 | 0.35 | 0.11 | vs. $D_{101}$ |
| 2013–2013 | $D_{101}$ (LKO) | MAC | 9 | 0.29 | 0.49 | 0.64 | 0.35 | vs. $D_{051}$ |
| 2013–2013 | $D_{051}$ (LKO) | MAC | 9 | 0.23 | 0.28 | 0.36 | 0.13 | vs. $D_{101}$ |
| 2014–2019 | $D_{101}$ (LKO) | AAC | 25 | 0.16 | 0.24 | 0.38 | 0.22 | vs. $D_{062}$ |
| 2014–2019 | $D_{062}$ (LKO) | AAC | 25 | 0.15 | 0.25 | 0.38 | 0.23 | vs. $D_{101}$ |
| 2014–2019 | $D_{101}$ (LKO) | AAC | 9 | 0.15 | 0.22 | 0.43 | 0.28 | vs. $D_{051}$ |
| 2014–2019 | $D_{051}$ (LKO) | AAC | 9 | 0.16 | 0.20 | 0.42 | 0.26 | vs. $D_{101}$ |
| 2014–2019 | $D_{101}$ (DAV) | AAC | 8 | 0.19 | 0.23 | 0.27 | 0.08 | vs. $D_{051}$ |
| 2014–2019 | $D_{051}$ (DAV) | AAC | 8 | 0.16 | 0.22 | 0.26 | 0.10 | vs. $D_{101}$ |
| 2014–2019 | $D_{051}$ (LKO) | AAC | 46 | 0.15 | 0.21 | 0.30 | 0.15 | vs. $D_{062}$ |
| 2014–2019 | $D_{062}$ (LKO) | AAC | 46 | 0.14 | 0.21 | 0.26 | 0.12 | vs. $D_{051}$ |
| 2016–2019 | $D_{101}$ (DAV) | AAD | 46 | 0.17 | 0.30 | 0.42 | 0.25 | vs. $D_{062}$ |
| 2016–2019 | $D_{062}$ (LKO) | AAD | 46 | 0.20 | 0.29 | 0.40 | 0.20 | vs. $D_{101}$ |
| 2016–2018 | $D_{101}$ (DAV) | AAD | 29 | 0.17 | 0.24 | 0.42 | 0.25 | vs. $D_{051}$ |
| 2016–2018 | $D_{051}$ (LKO) | AAD | 29 | 0.17 | 0.23 | 0.37 | 0.20 | vs. $D_{101}$ |
| 2019–2020 | $D_{051}$ (DAV) | AAD | 9 | 0.13 | 0.25 | 0.33 | 0.20 | vs. $D_{062}$ |
| 2019–2020 | $D_{062}$ (LKO) | AAD | 9 | 0.12 | 0.26 | 0.30 | 0.18 | vs. $D_{051}$ |

## 4.2 Period of manual vs. automated Dobson operation (MAC period)

Over the one and a half year period, while the data acquisition and measurement program for automatic operation were developed, different interventions interrupted and perturbed the measurements repeatedly. Changes in the automated operating procedures, their timing, and improvements of hardware components make the comparison between the systems challenging.

5 It was also demanding for the operators to measure continuously to get a sufficiently large data set of coincident measurements between the manual and automated instruments. During the Dobson inter-comparison campaign in July 2012, $D_{051}$ was also calibrated for ozone column measurements. Since March 2013, weather permitting, $D_{051}$ direct sun measurements have been recorded outside the higher priority Umkehr measurements periods. Therefore fewer coincident ozone column measurements of instruments $D_{101}$ and $D_{051}$ were recorded. Figure 6 presents the daily values of $\Delta_{101-062}$ (red) and $\Delta_{101-051}$ (black) in the

10 upper panel and the $\sigma_i$ values for the period 2012–2013 in the lower panel. The increase of the differences in summer 2013 suggests a drift of the Dobson $D_{101}$ instrument since the bias is similar for the two other instruments. In March 2013, a change





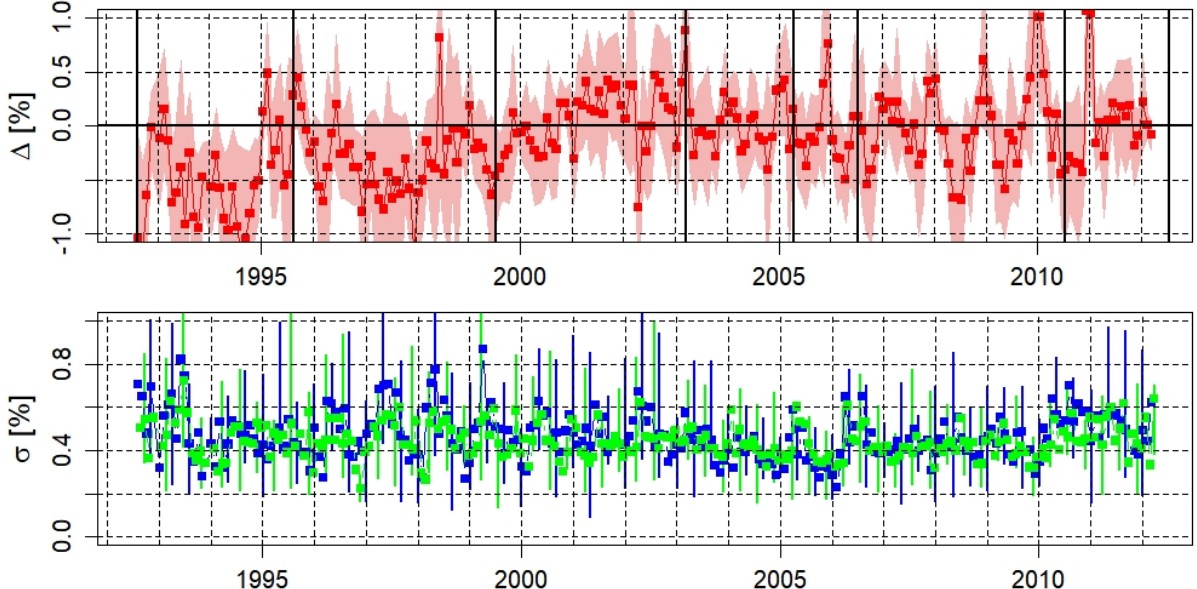

**Figure 5.** Daily analysis results: time series of the monthly median of the difference $\Delta_{101-062} = \delta_{D_{101}} - \delta_{D_{062}}$ (upper panel) and the individual $\sigma$ (lower panel) between coincident measurements of the Dobson instruments $D_{101}$ and $D_{062}$ over the period 1992–2012. Calibration campaigns are denoted by the black lines. The shading and the error bars are for the $IP_{2.5\%-97.5\%}$ interval.

of the azimuth control system was introduced but interference generated by this new system affected the measurements negatively. This problem was brought to light and solved in July 2013. In the first half of this MAC period, $D_{101}$ was $\sim$0.5% higher than $D_{062}$ and by mid-2013, the three instruments agree. The lower panel shows the improvement of the data quality with a significant decrease of the random fluctuations: the automated instruments (in red and orange) yield values around $\sim$0.3% while the manually operated instrument (in blue) is closer to $\sim$0.6%. In Figure 7, the monthly medians of $\Delta_{101-062}$, $\Delta_{101-051}$ and $\sigma_i$ are shown. With the exception of the period April–June 2013, the mean bias between the manual and automated instruments is within $\pm$0.6% and the repeatability of the automated Dobson is significantly reduced in comparison to the manually operated instrument.

Lines 2 and 3 in Table 3 show that $D_{101}$ data are on average 0.19% larger than $D_{062}$ data. However, we are looking at a bi-modal distribution due to the April–June 2013 period and the unevenly distributed measurements over the relatively short time period considered. Similarly, the negative value of $\Delta_{101-051}$=-0.56% is dominated by the spring 2013 period and the reduced sample of coincident measurements.

The 2012–2013 MAC comparison period shows that the agreement between manual and automated Dobson instruments is consistent and reproducible. Improved repeatability and the larger number of daily data are two of the prominent advantages of the automation.

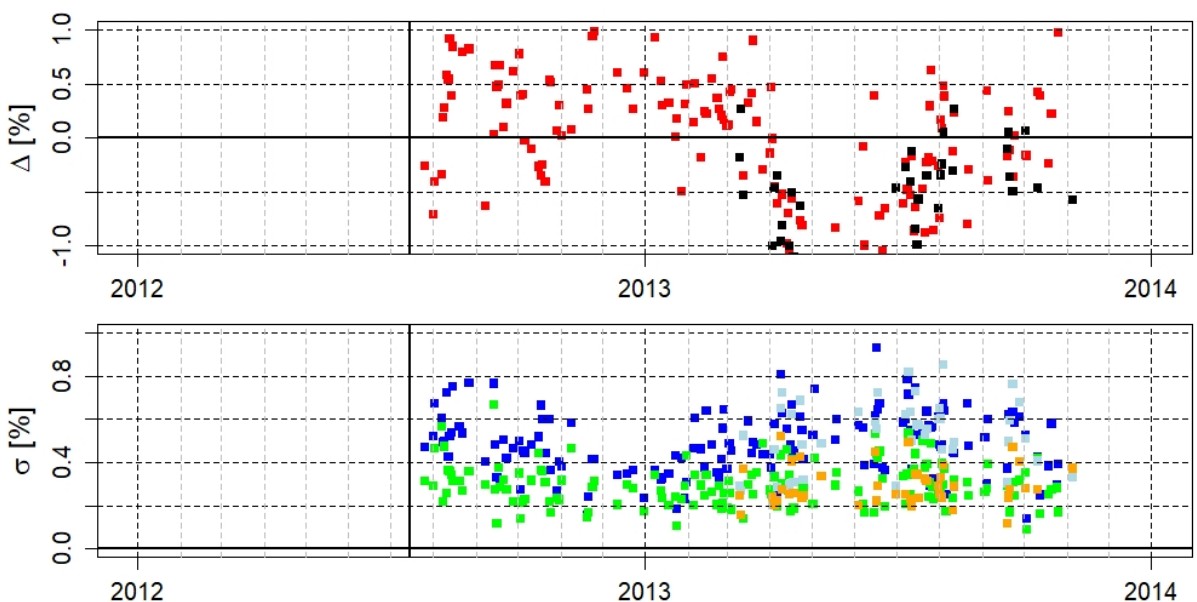

**Figure 6.** Upper panel: 2012–2013 time series of $\Delta_{101-062}$ (red) derived from coincident measurements of $D_{101}$ (manual) and $D_{062}$ (automated) and $\Delta_{101-051}$ derived from coincident measurements of $D_{101}$ (manual) and $D_{051}$ (automated) for 2013 (black). Lower panel: time series of $\sigma_i$ of $D_{101}$ (blue / light-blue), $D_{062}$ (green) and $D_{051}$ (orange).

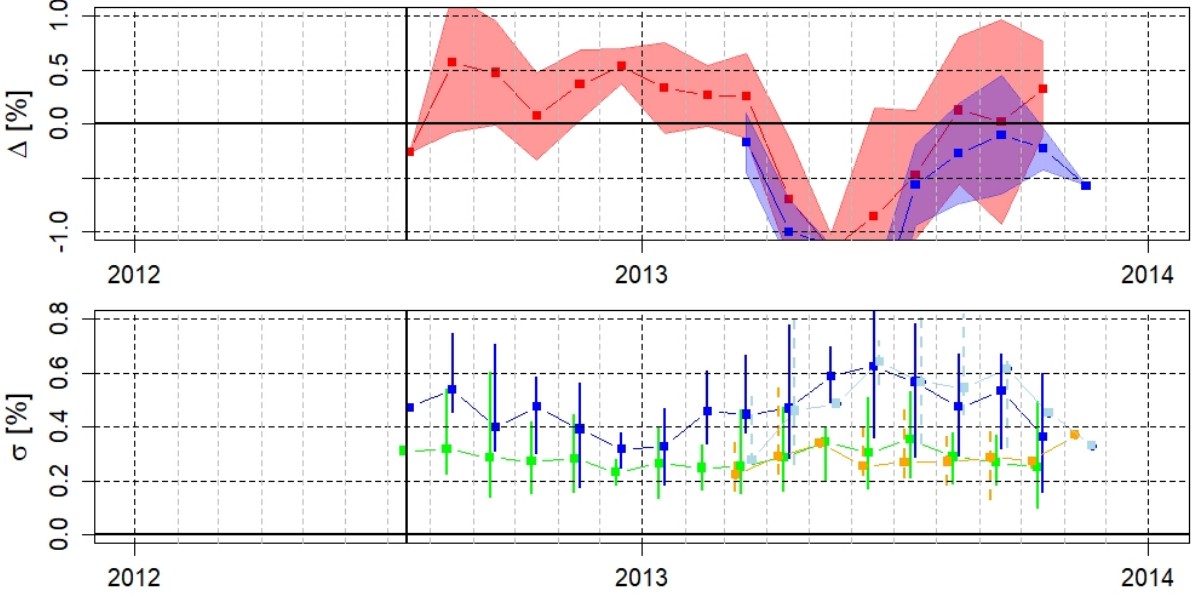

**Figure 7.** Upper panel: time series of the monthly median of $\Delta_{101-062}$ (red) and $\Delta_{101-051}$ (blue). Lower panel: time series of the monthly median of $\sigma_i$ of $D_{101}$ (blue / light-blue), $D_{062}$ (green) and $D_{051}$ (orange). The shading and the error bars correspond to the inter-percentile range $\text{IPR}_{2.5\%-97.5\%}$ of the various parameters.




### 4.3 Period of automated vs. automated collocated Dobson operation (AAC period)

$D_{101}$ automation was achieved by July 2014 and the period of intensive comparison between collocated automated Dobson instruments started. The data set for the pair of instruments $D_{101}$ and $D_{062}$ cover the period July 2014–December 2015 and a shorter period in 2017 and 2018. The second data set for the pair $D_{051}$ and $D_{062}$ covers the years 2013–2018. Since the relocation of $D_{051}$ at Davos at the end of 2018, $D_{101}$ and $D_{051}$ have been collocated there for a new AAC period. Similar to Figure 7, Figure 8 compares the automated measurements of the three pairs of collocated Dobson instruments $D_{101}$-$D_{062}$, $D_{101}$-$D_{051}$ and $D_{051}$-$D_{062}$. In March 2017, $D_{101}$ was back to Arosa after a transfer to Germany to characterize its slit function (*Köhler et al.*, 2018; *Stübi et al.*, 2020). Again for the July–August periods 2017 and 2018, $D_{101}$ was collocated with $D_{062}$ and $D_{051}$ in Arosa for calibration and maintenance campaigns. These transfers could have altered the instrument response but this is difficult to assess from these relatively short comparison periods. The monthly averages for these periods are also less representative since the sample is limited to only a few days in some cases. Notwithstanding, most data points lie within a ±1% interval with periods of lesser agreement. Overall, the period 2016–2018 shows a convergence of the differences in the ±0.5% range associated with the improvement and tuning of the Dobson instruments' control system. The time series of $\Delta_{101-062}$ (red strip in Figure 8) is mostly within the ±0.5% range except at the end of 2015 where $D_{062}$ seems to be slightly lower. The $\Delta_{051-062}$ (blue strip) shows the same deviation at the beginning of 2016 but converges to the ±0.5% range afterwards. The 2013–2014 period of the $\Delta_{051-062}$ time series indicates that the automated systems were not yet fully stable and that the bias could change by ±0.5% over a year time period. As shown in the AAC section of Table 3, the $\Delta_i$ for the difference pairs comparison are not significantly different from zero except for the pair $D_{101}$–$D_{051}$ at LKO. As evidenced in Figure 8, the 9 months between 2014–2018 mentioned in the table were not from a contiguous time period but typically reflect observations after $D_{101}$ displacements. Contrarily, in the 2019 period of collocation at Davos and after the 2018 calibration campaign, both instruments agree very well with an IPR$_{2.5\%-97.5\%}$ of less than ~1.3%. The lower panel of Figure 8 suggests a repeatability of the three Dobson instruments of around 0.2% with an IPR$_{2.5\%-97.5\%}$ of 0.1% to 0.5%. The values of $\sigma_i$ summarised in Table 4 (lines 5–12) are surprisingly similar for the lower 2.5%-percentiles (0.15–0.19 %) and the medians (0.20–0.25%) of total ozone values. The 97.5%-percentiles are on the order of 0.4%.

The results presented up to this point underline the stability and repeatability of the automated Dobson measurements.



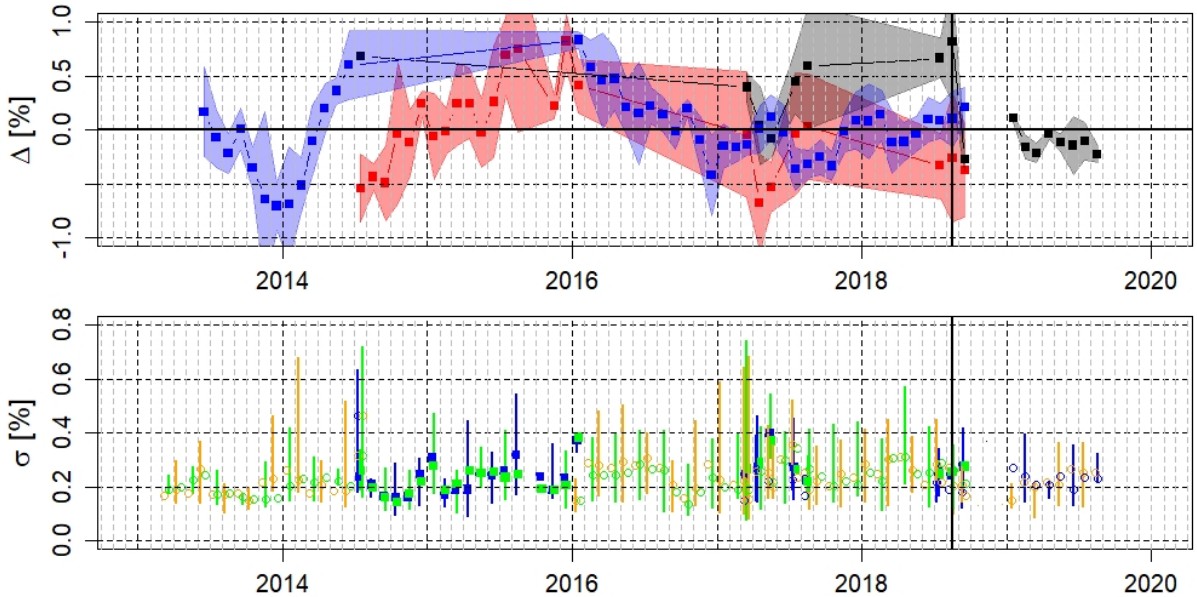

**Figure 8.** Upper panel: time series of the monthly median differences [%] between pairs of collocated Dobson instruments at Arosa over the period 2014–2019: $\Delta_{101-062}$ (red), $\Delta_{051-062}$ (blue) and $\Delta_{101-051}$ (black). Lower panel: time series of $\sigma_i$ monthly medians: $\sigma_{101}$ (blue), $\sigma_{062}$ (green), $\sigma_{051}$ (orange). The shading and the error bars (plot every two months for clarity) denote the inter-percentile range $IPR_{2.5\%-97.5\%}$.

### 4.4 Period of automated vs. automated distant Dobson operation 2016–2019 (AAD period)

In January 2016, the $D_{101}$ instrument was relocated to Davos with a set-up similar to the one at Arosa. Since September 2018, $D_{051}$ instrument has also been relocated to Davos. The line-of-sight distance between Arosa and Davos is 11 km. The sites are sufficiently close to suggest a similar large scale stratospheric ozone regime. However, the altitude difference between the two

observatories is 250 m which could translate into a slightly different total ozone column. Thus, total column ozone values at Davos are expected to be comparable or slightly larger than at Arosa. Since 2016, the data acquisition and computer controlled operation have had slightest changes compared to the previous period of developments. Similar to the previous Figures, Figure 9 compares the Dobson pairs in terms of $\Delta$ and $\sigma$ for the distant instruments. The $\Delta_{101-062}$ time series (red strip) is now mostly within 0.5%±0.5% which could be an indication of an average offset between the two stations of the order of ~0.5%. The most

recent data of 2019 tend to exhibit a smaller offset as also indicated by the $\Delta_{051-062}$ time series (blue strip). The $\Delta_{101-051}$ time series (black strip) has a very similar pattern which corroborates the agreement seen in Figure 8 between $D_{062}$ and $D_{051}$. Table 3 (lines 8–10) shows that the mean ozone column difference between Davos and Arosa is 0.53% ∈ [-0.30%, 1.05%] for the $D_{101}$-$D_{062}$ pair, 0.50% ∈ [-0.22%, 1.09%] for $D_{101}$-$D_{051}$ pair and 0.43% ∈ [-0.05%, 1.18%] for the $D_{051}$-$D_{062}$ pair. For the most recent 2019 period and after the 2018 calibration and maintenance campaign, the systematic differences are close

to ~0.4% which is in the range of 1–1.5 DU for the ozone column observed in the area. In the lower panel of Figure 9, the





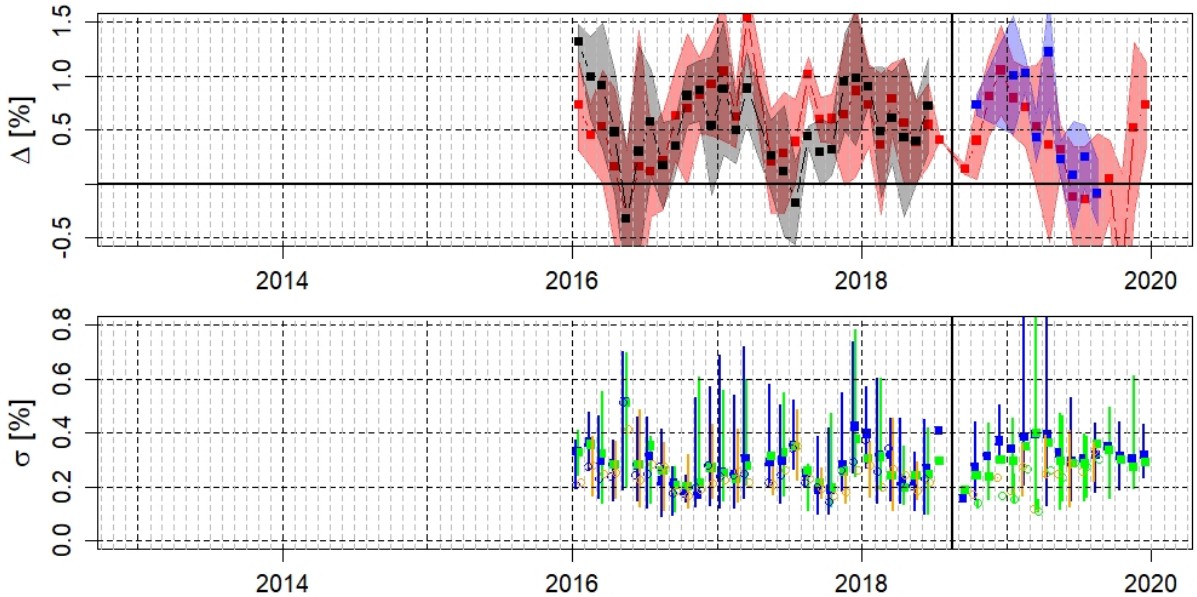

**Figure 9.** Upper panel: time series of the monthly median differences [%] between pairs of Dobson measurements at two different sites, $D_{062}$ at Arosa and $D_{101}$ / $D_{051}$ at Davos, over the period 2016–2020: $\Delta_{101-062}$ (red), $\Delta_{101-051}$ (black) and $\Delta_{051-062}$ (blue). Lower panel: time series of monthly medians of $\sigma_i$ : $\sigma_{101}$ (blue), $\sigma_{062}$ (green), $\sigma_{051}$ (orange). The shading and the error bars indicate the inter-percentile range $IPR_{2.5\%-97.5\%}$.

variations of $\sigma_i$ appear substantially larger than for the collocated cases. This is not too surprising since the two stations could certainly have different atmospheric conditions which influence the daily variations of the ozone column measured by the two distant instruments. In some cases, a time delay can be observed in the ozone variations at the two sites for example when a front is passing over the area (not shown). Attempts to systematically correct these time shifts did not improve the results

5 significantly so they were not implemented. The 97.5%-percentiles of the $\sigma$ of the Dobson instruments at different locations reached 0.6%–0.8% mostly in winter. Such values were less frequent in the case of collocated instruments (Figure 8). However, these observed larger $\sigma_i$ variations do not significantly affect the monthly averages in Table 4 for the AAD cases, which were in the range of 0.1%–0.4%.



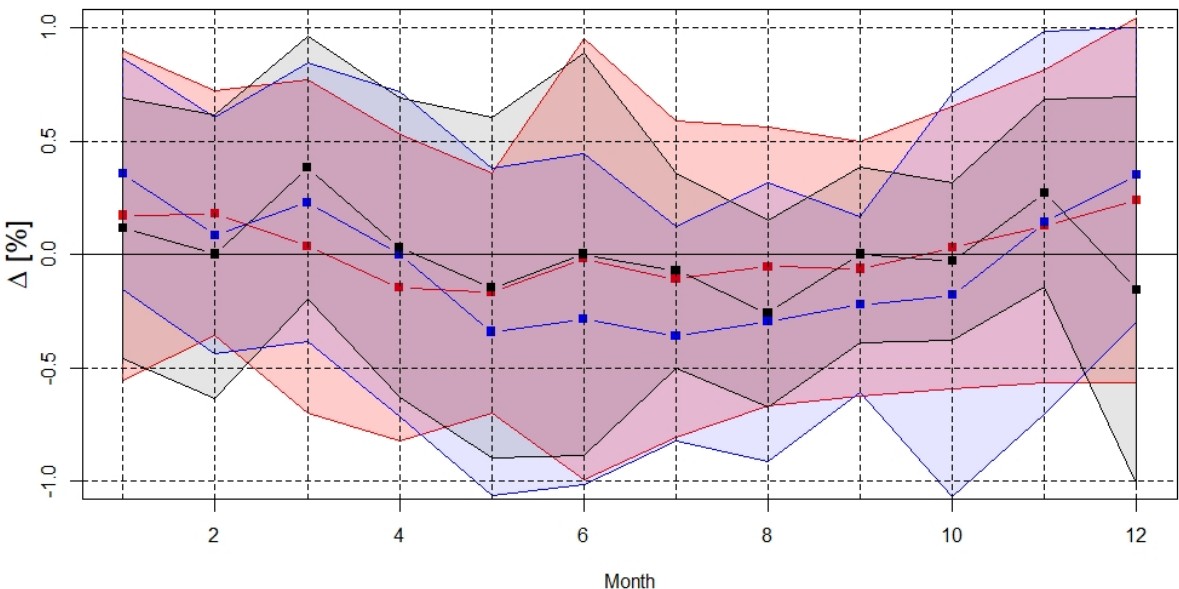

**Figure 10.** Annual cycle of the median differences of $\Delta_i[\%]$ : $\Delta_{101-062}$ (red) for sub-periods MMC; and $\Delta_{101-051}$ (black), $\Delta_{101-062}$ (blue) for sub-period AAD.

## 4.5 Seasonal cycle

For almost all optical measuring systems, a stray light effect is present with more or less influence on the measured values. The Dobson and Brewer sun spectrophotometers are no exception to this problem. The double-monochromator Brewer instruments are known to be free of a major stray light bias but the single-monochromator Brewer as well as the Dobson instruments are

affected (*Moeini et al.*, 2019; *Karppinen et al.*, 2015). The larger the ozone slant path (OSP = ozone amount $*$ air mass) the larger the stray light effect, because the signal at the shorter wavelengths decreases more rapidly and gets to the noise level. As the OSP is naturally seasonally dependant and the stray light effect is instrument dependant, it is of interest to analyse a possible bias due to the OSP. As noted in section 4.1, a seasonal cycle was observed in the 2005–2010 period and the right part of the upper panel of Figure 9 also shows a similar tendency. The result of the seasonal analysis of the $\Delta_i$ differences is

presented in Figure 10. The colored strips denoting the $IPR_{2.5\%-97.5\%}$ largely cover the zero line but the medians show a trend with negative values in summer and positive values in winter. This is more pronounced for the AAD cases of the $D_{101}$–$D_{062}$ pair (in blue) where the amplitude of $\sim0.8\%$ is twice as large as for the MMC cases (in red). The $D_{101}$–$D_{051}$ pair exhibits monthly differences that are more random. The 2018 intercomparison revealed an OSP-dependent bias between the European Dobson traveling standard $D_{064}$ and both $D_{101}$ and $D_{051}$, but showed no such bias between $D_{064}$ and $D_{062}$. Without being

firmly conclusive, the seasonal analysis suggests a possible contribution of a stray light induced bias caused by $D_{101}$ and/or $D_{051}$ instrument.





## 5  Discussion

In section 4, the analysis of coincident measurements of three Dobson instruments from Arosa LKO is presented for four differ-
ent configurations named MMC, MAC, AAC and AAD that refer to manual (M) or automatic (A), collocated (C) or distant (D)
operation of the instruments. The method used to separate the mid- to long-term systematic biases between instruments and the
short term random variations associated with each instrument were first presented in *Stübi et al.* (2017a). This method allowed
us to reduce by half the overall global bias range from typically $IPR_{2.5\%-97.5\%}$ ∼3% (Table 2) down to $IPR_{2.5\%-97.5\%}$ ∼1.5%
(Table 3).

The 20 year MMC period was long enough to bring both Dobson instruments $D_{101}$ and $D_{062}$ in agreement after multiple
calibration campaigns. No significant biases were observed within the uncertainty associated with manual operations and the
rather limited number of daily observations. Dobson instrument $D_{051}$ was primarily dedicated to automated Umkehr measure-
ments, which makes direct sun observations very difficult.

The development of the Dobson automation from scratch took a few years with periods of hardware and software changes
that impacted on the measurement stability. For back-up measurements, $D_{101}$ continued to be manually operated in parallel to
the automated $D_{062}$ and $D_{051}$ over the 2012–2013 period. The analysis of the relatively short (1.5 years) MAC period shows
an overall good agreement albeit with sub-periods of biased measurements due to a malfunctioning of the automated system.
Notwithstanding, the automated instruments proved to perform equally well or even better than a manually operated instru-
ment. Dobson $D_{051}$, which was newly used also for direct sun observations of the ozone column during the MAC period,
yields larger ozone values compared to the manual Dobson $D_{101}$ instrument. As shown in Figures 6 and 7, the $\Delta_{ij}$ values of
the $D_{101}-D_{062}$ and $D_{101}-D_{051}$ instrument pairs are similar, as are the $\sigma_i$ values. In Table 2, the mean differences for the MAC
cases are not significant considering the large $IPR_{2.5\%-97.5\%}$ ∼ 3.8% for the coincident $D_{101}- D_{062}$ and $D_{101}-D_{051}$ values.
From the refined daily analysis, the $IPR_{2.5\%-97.5\%}$ have been reduced to ∼ 1.6% (Table 3). Even though the values of the
differences for the two pairs appear to be quite different ($\Delta_{101-062} = 0.19\%$ vs. $\Delta_{101-051} = $ -0.56%), they still remain close to
±0.5%. Moreover, they represent averages of different time periods and sample lengths and should be compared with caution.

Beginning in 2014, all three Dobson instruments were ready for automated and collocated (AAC) operation. For a while, as
shown in Table 1, the operating environment was still changing from time to time, and the system was subject to occasional
technical glitches. Table 2 shows that the direct comparison differences for the AAC case are not significant with ten times
larger sample sizes than in the MAC case. The daily analysis results from Table 3 confirm the excellent agreement between
$D_{062}$ and both $D_{101}$ and $D_{051}$ while the pair $D_{101}$ and $D_{051}$ presents a barely significant value of $\Delta_{101-051} = 0.45\% \in$ [-0.23%,
0.79%]. As mentioned in section 4.3, the $D_{101}-D_{051}$ coincident data sets were recorded for three distinct periods (black sym-
bols on Figure 8) with reduced sample sizes and are therefore less representative. The recent 2018–2019 period of coincident
measurements at Davos with a value of $\Delta_{101-051} = $ -0.13% ∈ [-0.22%, 0.08%] confirms the excellent agreement between the
$D_{101}$ and $D_{051}$ instruments. In summary, the automated Dobson systems were very reliable and reproducible during this AAC
comparison period.

Considering the homogeneity and continuity of the Arosa / Davos ozone column time series, the comparison of coincident





data obtained independently at the two stations is an essential part of this study. A similar analysis by *Stübi et al.* (2017b) considering the long term stability and random uncertainties of the Brewer instruments found no significant differences between the Arosa and Davos sites. The analysis of the AAD period presented in section 4.4 arrives at the same conclusion. Notwithstanding, the last three lines in Table 2 may indicate the possibility of a $\simeq 0.4\%$ systematic high bias within an $IPR_{2.5\%-97.5\%}$

of 2.5–2.9 for the instruments located at Davos. The daily analysis results in Table 3 confirm these numbers with $\Delta_i$ values of $\simeq 0.5\%$ but with a reduced $IPR_{2.5\%-97.5\%}$ of $\sim 1.3\%$. In *Stübi et al.* (2017b), the authors estimated that the Arosa–Davos altitude difference of 260 m could contribute $0.25\% \pm 0.15\%$ to the ozone column. Therefore half of the observed difference could be attributed to the longer ozone column measured from Davos. The $\sigma_i$ values reported in Table 4 are consistent and demonstrate the benefits of automation. Manual operation of the Dobson instrument yields values of $\sigma \simeq 0.40\% \in [0.3\%,$

$0.7\%]$ on average. The automation of the operations reduced these values to $\sigma \simeq 0.25\% \in [0.15\%, 0.40\%]$. These numbers are slightly lower than the corresponding Brewer values of around 0.3%–0.4% reported in *Stübi et al.* (2017a)(table 3) and *León-Luis et al.* (2018)(table 4). These values further confirm the good quality of the automated Dobson measurements.

The slight seasonal component presented in section 4.5 is probably responsible for the ripples observed in Figures 8 and 9. Even though all Dobson instruments are based on a similar design, the stray light bias is instrument dependent. An improved

processing algorithm including stray light correction as presented in *Moeini et al.* (2019) could be applied for the Arosa-Davos data since Brewer double monochromator instruments are collocated. Recent slit function measurements of the Arosa Dobson instruments are now available from the ATMOZ project (*ATMOZ*, 2018). However, such improvements were beyond the scope of the present analysis. Similarly, the characterisation of the first few kilometers of the ozone profile and its seasonal cycle in the Arosa and Davos valleys, to more accurately assess differences of the free troposphere ozone column above these two sites,

needs to be refered to future research.

The present results based on Dobson data confirm the conclusion reported in *Stübi et al.* (2017b) based on Brewer data. Biases found are not statistically significant at the $IPR_{2.5\%-97.5\%}$ level, and therefore, could not be systematically compensated. A re-processing of the Dobson and Brewer data sets with an improved algorithm based on recent ozone cross-section values, improved stray-light correction based on better slit functions could perhaps reduce the uncertainties on the biases found but

would most certainly not change our conclusions. The results presented in this study are unique since no other station of the Dobson network has operated fully automated collocated Dobson instruments over a multi-years time period. Considering the importance of the Arosa time series, research will continue with a focus on trend analyses and break detection of the series both on data from Arosa (continued until mid 2021) and data based on the combined Arosa-Davos time series.



# 6 Data availability.

The data used for this analysis are available at the WOUDC for the Dobson $D_{101}$ (1992–2014) and $D_{062}$ (2014–2020) instruments. The complete data sets can be requested by direct contact with the corresponding author.

*Author contributions.* R. Stübi was mainly responsible for the data analysis and the first version of the manuscript. H. Schill was in charge
5 of the quality control and the preparation of the data sets. J. Klausen, E. Maillard Barras and A. Haefele contributed to the data interpretation
and revisions of the manuscript.

*Competing interests.* No competing interests.

*Acknowledgements.* We would like to thank the PMOD/WRC staff for their great support to run our instruments in their premises and for
the excellent collaboration.





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
