# Peer review of "Quality assessment of Dobson spectrophotometers for ozone column measurements before and after automation at Arosa and Davos"

_Atmospheric Measurement Techniques, 2020_

## Referee Comment (RC1) · Anonymous Referee #1 · 24 Dec 2020

Quality assessment of Dobson spectrophotometers for ozone column measurements before and after automation at Arosa and Davos. René Stübi, Herbert Schill, Eliane Maillard Barras, Jörg Klausen, and Alexander Haefele

Initial Comments: I found this manuscript, describing the careful updating of several traditional manually operated Dobson Ozone Spectrophotometers and relocation of the instruments to a new site without breaking the long historical measure series, to be very detailed and complete. The manuscript fits AMT's mission as it is a commentary on both laboratory measurement technique and an observational program. I recommend publication of this manuscript after minor technical corrections below.

[Figure]

Specific Comments:

3.1 Data quality control: I like the detail in the explanation of the automated procedure. I would have liked to have seen a daily record before and after this procedure was applied.

Technical Corrections/Comments/Suggestions General: The use of the subscripts to identify specific Dobson instruments is not consistent. For example: D062 vs D062 Page 2, Line 5: Suggest: However, while ozone layer depletion seems to have stopped since the beginning of the 21st century, the expected recovery of the ozone layer to the pre-1980 level has still not been observed in most parts of the atmosphere. Page 2, Line 31. The first mention of the term Umkehr should have a reference or explanation that this is a measurement designed to determine an ozone profile with height. Page 4, Line 15: Suggest explaining that the optical alignment of Dobson instruments (Dobsons) is standardized for all instruments. Table 1, Line 3: Define SOOH. Line 9: Define MOHp. Line 15: Define SL- (Only place in manuscript that these terms are used) Table 2, Caption: Suggest defining the specific standard instrument. Figure 3, Consider using a different color, and line weight for the arrows indicating the calibration / maintenance campaign for better visibility. Page 23, Line 6, Komhyr (spelling) and Page 3 Line 8 Page 23, Line 28: Citation should be: León-Luis, S. F., Redondas, A., Carreño, V., López-Solano, J., Berjón, A., Hernández-Cruz, B., and Santana-Díaz, D.: Internal consistency of the Regional Brewer Calibration Centre for Europe triad during the period 2005–2016, Atmos. Meas. Tech., 11, 4059–4072, https://doi.org/10.5194/amt-11-4059-2018, 2018.

---

## Referee Comment (RC2) · Anonymous Referee #2 · 28 Dec 2020

General comments:

- The content of the paper is a very interesting and important contribution to the issue of data quality in the Dobson spectrophotometer network and its improvement. The content and structure of the publication is very complex and it is sometimes not easy to follow the presentation of the results and to understand the presented differences in the various scenarios. Its objective fits in any case the AMT requirements and therefore it is important and worth being published with some minor, few major and some technical corrections.

Specific comments:

i. Minor Issues:

- P1: The results of the AAD-scenario is not addressed in the Abstract

- Introduction P2 L2: not only use is banned, but production too

- The slight decrease of variability (p2 l13) is not obvious in figure 2, and if it really exists it might have been caused by atmospheric reasons too.

- P2 LL16 – 19: It would be good to explain, that the already seen recovery in high altitudes is driven by chemical reasons, and the still existing ambiguity in the lower stratosphere probably comes from dynamical effects (due to climatic change?)

- P3 L11: wavelengths range starts below 310 nm (Ashort = 305.5 nm)

- P4 L8: logarithmic differences

- P4 LL20 – 23: it is correct, that the application of effective slit functions / absorption coefficients on the historical data will not be possible, but what about the effective temperature of the ozone layer. Might there be a chance for such a correction?

- P4 LL24 – 27: the Dobson calibration system consists of 1 World Dobson Calibration Center WDCC (with 1 primary standard Dobson and one traveling primary standard Dobson) and 5 Regional Dobson Calibrations Centers RDCC (with 6 secondary standard Dobsons, among them 2 in Europe)

- P 5, table 1: the listing of Dobsons ICs after 2012 is not complete and what about the installation of new electronics in 2005/2006 (D051 in 2006 and not in 2011?)

- P6, figure2: European regional standard instead of travelling

- Table 2 and 3: headlines for the second and third column should be identical: reference Dobson and redundant Dobson

- What is the difference between D101 (blue/light-blue) in figures 6 and 7?

- P18 L10: the term trend is not a correct one

ii. Major issues:

- Introduction P2: it is not mentioned (e.g. in l10) that the Arosa long term Dobson record started in 1926, the earliest mentioned date is 1948. Figure 1 clearly shows this early start

- The leveling off (P2 L11 and in figure 1) starts obviously already at the end of the nineties and not in 21st century

- It is generally a pity, that the relocation of the observations from Arosa to Davos is explained only in a short statement (p3 l1), as this is very important to understand, that the Davos record will be appropriate to continue the famous Arosa record. It should be mentioned already here, that corresponding investigations are planned to confirm the homogeneity of a combined Arosa/Davos record, as it is done later under Discussion on pp19/20.

- P9 and P11, tables 2 and 3: I have problems to understand, why some of the signs of the median values of the differences are reversed. If this comes from different calculation methods (differences of coincident data and of polynomial fit) it should be explained. Moreover the mentioned time periods in the text and in the two tables (AAD 2016 – 2019 and AAD 2014 – 2019, respectively) are not consistent and therefore confusing

- P11 text and P13, Figure 5: The statement, that calibration campaigns did not induce noticeable breaks is in contrast to published reports. The campaign in 1999 revealed (official report GAW No. 138) that D101 was more than 1% too low, whereas D062 was less than 1% too low. Thus D101 was corrected, but not D062.

- Some of the figures (6, 7, 8 and 9) truncate shaded areas and/or curves. The y-axes should adjusted correspondingly to avoid truncation

- Figure 8: The continuous shading of time periods with gaps (blue and red) is not consistent (black is ínterrupted); moreover the colours black and dark blue can hardly

be distinguished.

- PP16-17 and figure 9: Is there any explanation, why the 2019-differences are close to zero for some months

- P18, section 4.5: can the statements of OSP-depending biases, revealed during the 2018 intercomparison be confirmed by some graphs or so? Are there differences between initial and final data (improvements?)?

Technical corrections:

- P2 L9: German instead of german and Lichtklimatisches instead of Lichtklimatsches

- P4 L16 and later in P20 L17: ATMOZ 2018 is cited, but 2018 is missing in the reference list

- There are two cited publication Stübi et al 2017 on different pages, they should be different 2017a and 2017b.

- P11, L9: panel instead of panle

- P15 L3: covers instead cover

- References, P23: Sergio Fabian Leon-Luis should be Leon-Luis, Sergio Fabian as it is cited in the text; moreover the year of publication of the SPARC/IO3C/GAW-report should be set to end to be consistent with the other references.

---

## Author Comment (AC1) · 11 Mar 2021

Reply to referee#1 comments on AMT-2020-441 manuscript, "Quality assessment of Dobson spectrophotometers for ozone column measurements before and after automation at Arosa and Davos" by René Stübi et al."

The authors thank referee#1 for the valuable comments and suggestions that allow us to improve our manuscript.

Comment 1.  3.1 Data quality control: I like the detail in the explanation of the automated procedure.  I would have liked to have seen a daily record before and after this

procedure was applied.

Reply: It would have been difficult to illustrate the procedure with a single figure since many atypical cases regarding the outliers are of interest. The selection of the parameters (polynomial function, thresholds) were empirically determined by trial and error. An example (as illustrated in Figure 1) would have been possible but it does not show the variety and complexity of outliers' detection.

Comment 2. The use of the subscripts to identify specific Dobson instruments is not consistent.

Reply: We re-established the consistency of the various Dobson subscripts.

Comment 3. Page 2, line 5, suggestion of a different sentence

Reply: We adopted the proposed wording.

Comment 4. Page 2, Line 31.The first mention of the term Umkehr should have a reference or explanation.

Reply: We introduced a sentence to explain the Umkehr method and a reference to Petropavlovskikh et al. (2009).

Comment 5. Page 4, Line 15: Suggest explaining that the optical alignment of Dobson instruments (Dobsons) is standardized for all instruments.

Reply: We introduced a sentence along the lines suggested by the referee.

Comment 6. Table 1, Line 3: Define SOOH. Line 9: Define MOHp.

Reply: A new paragraph explaining the WMO Dobson calibration procedures is introduced in response to a comment of the second referee. The acronyms SOOH and MOHp are therefore defined in the new text.

Comment 7. Line 15: Define SL- (Only place in manuscript that these terms are used)

Reply: We have removed the abbreviation SL and given the full words.

Comment 8. Table 2, Caption: Suggest defining the specific standard instrument.

Reply: This suggestion is difficult to satisfy since there is no standard instrument. We stated D101 as the reference during the MMC and MAC when it is present in the pair comparison. However, we do not have an instrument that is more "standard" than the others are.

Comment 9. Figure 3, Consider using a different color, and line weight for the arrows indicating the calibration / maintenance campaign for better visibility.

Reply: We increased visibility of the arrows.

Comment 10. Page 23, Line 6, Komhyr (spelling) and Page 3 Line 8

Reply: Spelling corrected

Comment 11. Page 23, Line 28: Citation should be: León-Luis, S. F., Redondas, A., Carreño, V., López-Solano, J., Berjón, A., Hernández-Cruz, B., and Santana-Díaz, D.: Internal consistency of the Regional Brewer Calibration Centre for Europe triad during the period 2005–2016, Atmos. Meas. Tech., 11, 4059–4072, https://doi.org/10.5194/amt-11-4059-2018, 2018.

Reply: Reference corrected.
* * *
[Figure]

[Figure]

**Fig. 1.** Figure outliers_removal

---

## Author Comment (AC2) · 11 Mar 2021

Reply to referee#2 comments, AMT-2020-441 manuscript, "Quality assessment of Dobson spectrophotometers for ozone column measurements before and after automation at Arosa and Davos" by René Stübi et al."

The authors thank referee#2 for the critical reading and the valuable comments and suggestions that allow us to improve our manuscript.

Referee#2 comments and authors Reply : follow :

i. Minor Issues:

[Figure]

Comment 1. P1: The results of the AAD-scenario is not addressed in the Abstract.

Reply : Sentence added

Comment 2. Introduction P2 L2: not only use is banned, but production too

Reply : Comment added

Comment 3. The slight decrease of variability (p2 l13) is not obvious in figure 2, and if it really exists it might have been caused by atmospheric reasons too.

Reply : Effectively, a smaller variability seems a transient phenomenon in the 90ies (Pinatubo consequences?). Sentence removed.

Comment 4. P2 LL16 – 19: It would be good to explain, that the already seen recovery in high altitudes is driven by chemical reasons, and the still existing ambiguity in the lower stratosphere probably comes from dynamical effects (due to climatic change?)

Reply : The chemistry vs. dynamical changes are inserted as suggested.

Comment 5. P3 L11: wavelengths range starts below 310 nm (Ashort = 305.5 nm)

Reply : Corrected range: 305-340 nm

Comment 6. P4 L8: logarithmic differences

Reply : Corrected

Comment 7. P4 LL20 – 23: it is correct, that the application of effective slit functions / absorption coefficients on the historical data will not be possible, but what about the effective temperature of the ozone layer. Might there be a chance for such a correction?

Reply : Effectively, a climatology of the effective temperature based on earlier datasets (Aerological sounding since 40s, Umkehr since 1957, ozone soundings since1966, reanalysis) would certainly correct a major part of this effect. This work is planned in a reprocessing of the Arosa series. The word "impossible" is changed to "harder".

Comment 8. P4 LL24 – 27: the Dobson calibration system consists of 1 World Dobson Calibration Center WDCC (with 1 primary standard Dobson and one traveling primary standard Dobson) and 5 Regional Dobson Calibrations Centers RDCC (with 6 secondary standard Dobsons, among them 2 in Europe)

Reply : A new paragraph describing of the Dobson network calibration process is added as suggested.

Comment 9. P 5, table 1: the listing of Dobsons ICs after 2012 is not complete and what about the installation of new electronics in 2005/2006 (D051 in 2006 and not in 2011?)

Reply : The table is not exhaustive but remind some major events. However, the selection of the events was not optimal as pointed out by the referee. We decided to mention only the event of the transition period 2010-2015 and adapted the table accordingly. The new electronic changes from 2005/2006 were made on D062 and D101 while D051 electronic were adapted in 2011.

Comment 10. P6, figure2: European regional standard instead of travelling

Reply : Corrected

Comment 11. Table 2 and 3: headlines for the second and third column should be identical: reference Dobson and redundant Dobson

Reply : Corrected

Comment 12. What is the difference between D101 (blue/light-blue) in figures 6 and 7?

Reply : D101 data are compared independently to D062 and to D051 data resulting in slightly different ïĄş for D101ïĂő For the pair D101-D062, ïĄş symbols are in blue and in light-blue for the pair D101-D051.

Comment 13. P18 L10: the term trend is not a correct one

[Figure]

Reply : Effectively, trend refers commonly to a linear type of behavior so the term "curvature" has replaced "trend".

ii. Major issues:

Comment 14. Introduction P2: it is not mentioned (e.g. in l10) that the Arosa long term Dobson record started in 1926, the earliest mentioned date is 1948. Figure 1 clearly shows this early start

Reply : Reference to the 1926 early measurements has been added.

Comment 15. The leveling off (P2 L11 and in figure 1) starts obviously already at the end of the nineties and not in 21st century

Reply : Corrected to mid-90s

Comment 16. It is generally a pity, that the relocation of the observations from Arosa to Davos is explained only in a short statement (p3 l1), as this is very important to understand, that the Davos record will be appropriate to continue the famous Arosa record. It should be mentioned already here, that corresponding investigations are planned to confirm the homogeneity of a combined Arosa/Davos record, as it is done later under Discussion on pp19/20.

Reply : A new paragraph mentioning the GCOS principles has been added. The adherence to these principles and the present analysis validates the continuity of the Arosa/Davos series which, anyhow will be carefully checked in the future with additional data.

Comment 17. P9 and P11, tables 2 and 3: I have problems to understand, why some of the signs of the median values of the differences are reversed. If this comes from different calculation methods (differences of coincident data and of polynomial fit) it should be explained. Moreover the mentioned time periods in the text and in the two tables (AAD 2016 – 2019 and AAD 2014 – 2019, respectively) are not consistent and therefore confusing.

Reply : The confusion was due to the exchange of "reference series" vs. "redundant series" in the definition of the differences. It has been homogenized in the tables and figures to remove the confusion. Similarly, the time periods have been homogenized.

Comment 18. P11 text and P13, Figure 5: The statement, that calibration campaigns did not induce noticeable breaks is in contrast to published reports. The campaign in 1999 revealed (official report GAW No. 138) that D101 was more than 1% too low, whereas D062 was less than 1% too low. Thus D101 was corrected, but not D062.

Reply : The comment is correct but the reprocessing of the series backward in time (based on the lamps tests results) lessen pre-existing differences.

Comment 19. Some of the figures (6, 7, 8 and 9) truncate shaded areas and/or curves. The y-axes should adjusted correspondingly to avoid truncation

Reply : The figures scales will be adjusted in the final version.

Comment 20. Figure 8: The continuous shading of time periods with gaps (blue and red) is not consistent (black is ínterrupted); moreover the colors black and dark blue can hardly be distinguished.

Reply : The long (black) shading was interrupted for clarity but it will be re-introduced for the final version and the color adapted.

Comment 21. PP16-17 and figure 9: Is there any explanation, why the 2019-differences are close to zero for some months

Reply : There is no clear reason but lower values appeared also in summer 2016 and 2017 that point toward a small seasonal cycle of the differences possibly related to OSP.

Comment 22. P18, section 4.5: can the statements of OSP-depending biases, revealed during the 2018 intercomparison be confirmed by some graphs or so? Are there differences between initial and final data (improvements?)?

[Figure]

Reply : No figure of the time series around 2018 intercomparison has been done along the line suggested in comment 22. In the follow up analysis with longer data set from Davos, such analysis would be of interest especially characterizing the stray light of the Dobson with parallel measurements of the double Brewer.

Technical corrections:

Comment 23. P2 L9: German instead of german and Lichtklimatisches instead of Lichtklimatsches

Reply : corrected

Comment 24. P4 L16 and later in P20 L17: ATMOZ 2018 is cited, but 2018 is missing in the reference list

Reply : corrected

Comment 25. There are two cited publication Stübi et al 2017 on different pages, they should be different 2017a and 2017b.

Reply : corrected

Comment 26. P11, L9: panel instead of panle

Reply : corrected

Comment 27. P15 L3: covers instead cover

Reply : corrected

Comment 28. References, P23: Sergio Fabian Leon-Luis should be Leon-Luis, Sergio Fabian as it is cited in the text; moreover the year of publication of the SPARC/IO3C/GAW-report should be set to end to be consistent with the other references.

Reply : corrected